# Real-Time Locating System in Production Management

**DOI:** 10.3390/s20236766

**Published:** 2020-11-26

**Authors:** András Rácz-Szabó, Tamás Ruppert, László Bántay, Andreas Löcklin, László Jakab, János Abonyi

**Affiliations:** 1MTA-PE Lendület Complex Systems Monitoring Research Group, Department of Process Engineering, University of Pannonia, Egyetem u., 10, POB 158, H-8200 Veszprém, Hungary; andrasraczszabo91@gmail.com (A.R.-S.); laszlo.bantay@gmail.com (L.B.); janos@abonyilab.com (J.A.); 2Sunstone-RTLS Ltd., Kevehaza u., 1-3, H-1115 Budapest, Hungary; support@sunstone-rtls.com; 3Institute of Industrial Automation and Software Engineering, University of Stuttgart, Pfaffenwaldring 47, D-70550 Stuttgart, Germany; andreas.loecklin@ias.uni-stuttgart.de

**Keywords:** RTLS, indoor positioning system (IPS), position data, industry 4.0, traceability, product tracking

## Abstract

Real-time monitoring and optimization of production and logistics processes significantly improve the efficiency of production systems. Advanced production management solutions require real-time information about the status of products, production, and resources. As real-time locating systems (also referred to as indoor positioning systems) can enrich the available information, these systems started to gain attention in industrial environments in recent years. This paper provides a review of the possible technologies and applications related to production control and logistics, quality management, safety, and efficiency monitoring. This work also provides a workflow to clarify the steps of a typical real-time locating system project, including the cleaning, pre-processing, and analysis of the data to provide a guideline and reference for research and development of indoor positioning-based manufacturing solutions.

## 1. Introduction

Getting accurate and actual information of a process status is very important in the management and development of production systems. Information is often position located; this way, it defines the actual position of a workpiece or resource in the production area. This location based information may be suitable to connect information of resources and activities/workpieces. The purpose of this article is to introduce the potential in tools developed for indoor positioning, as well as the available technologies and the possible use of data hidden in information.

According to the ISO/IEC 24730-1:2014 standard, the real-time locating system (RTLS) is a wireless system used to locate the position of an item anywhere in a defined space at a point in time that is or is close to real-time. Indoors positioning systems (IPS) [1] locate objects in closed structures, such as office buildings, hospitals, stores, factories, and warehouses, where the GPS proves to be inaccurate [2]. In this paper we focus on how indoor positioning can be utilized in manufacturing and for simplicity, we refer to these indoor positioning systems as RTLS.

Several surveys and comparative analyses can be found on indoor tracking technology based on localization techniques [3,4,5]. One article has attempted to classify techniques and systems by presenting a comprehensive performance comparison of the accuracy, precision, complexity, scalability, robustness and cost [6]. Similarly, studies comparing RTLS technologies can also be found in the literature [7,8,9,10,11,12,13]. Furthermore, a meta-review provides a comprehensive compilation of 62 survey papers on the topic of RTLS [14], and the classification of current typical RTLS is introduced with a layered conceptual framework [15]. Many technologies are available such as infrared light, ultrasound, laser and their combinations. The reviews of these technologies focus on the technical elements and standalone applications and show that only a few specific industrial applications are available [16].

This paper aims to provide an overview of the applicability of RTLS in manufacturing to support the practical applications and provide a guideline or reference for implementation, research and development of indoor positioning and RTLS.

In order to explore the potential applications a systematic examination of literature was performed in Scopus, following the PRISMA-P protocol. The used keyword set (“real-time positioning systems” OR “indoor positioning systems”) AND (“manufacturing” OR “industry”) resulted near to 300 articles from which the thematic groups of the related research were identified.

Positioning data in the production system is the key information for traceability [17], and digitalization [18]. We overview the potential technologies and the possible traceability levels in Section 2. The levels represent the identification unit from the transportation unit (highest level— trucks, ships) to item unit (lowest level–raw material). Determination of the traceability level depends on more factors such as the complexity of the production process, the number of raw material types and the conditions of the information system and infrastructure.

We discuss the potential manufacturing applications according to the tasks depicted in Figure 1. The figure describes how positioning-based information allows for continuous improvement to other parts of the manufacturing environment, such as production control, logistics, applications in quality management, safety and RTLS-based efficiency monitoring. These applications and the required data analysis tasks are discussed in Section 3.

Section 4 describes a workflow to implement an RTLS-based digitalization project such as installation with the necessary hardware elements and data processing to allow the data integration. Finally, the applicability of the RTLS in manufacturing is illustrated by a case study presented in Section 5.

## 2. Levels of Location Information in Manufacturing Industries

To determine the appropriate tracking technology, we need to know the identification levels with the associated technologies. In terms of integration into our system, it is important to see the relevant characteristics for the selection process of the particular RTLS technology. In the next subsection, we present the criteria for choosing a method for a tracking solution that arises in a production system. Based on this information in the second subsection, we show recommended indoor positioning-based traceability technologies. Please note that a detailed description of the technologies is not the purpose of this article.

### 2.1. Identification Levels and Technology Solutions

Different applications require different types of tracking systems. Figure 2 shows the identification layers based on the possible available levels.

GPS is used for tracking containers and transportation equipment. At the lower level, where we consider the intralogistics, GPS is not accurate, or in many cases, it is unusable and not suitable for general asset tracking due to energy consumption. RTLS can handle indoor container identification as well as the unit load, transportation unit and package, that is, the third, second and first layers. RFID and barcode technologies are possible solutions for item identification, but it is essential to consider whether using a particular technology is appropriate or if it is worthwhile to combine the available technologies, such as UWB technology (which is not recommended for raw material tracking). This approach is called hybrid traceability technology in the literature [19]. However, with RFID tags, unit identification can be achieved with a lower cost [7].

Table 1 shows solutions for these different identification levels (see in Figure 2) with the advantages and disadvantages. We discuss four different traceability solutions to support the technology chosen. A decision-making model for selection is proposed in Reference [16], where UWB, RFID, Wifi, Zigbee and BLE (Bluetooth Low Energy) are compared with several aspects based on the developed methodology. The steps of this methodology are RTLS definition, market analyses, weights of criteria, ranking. They made a comparison between the technologies with the many parameters (coverage area, accuracy, room level usefulness, RF interference potential, bit rate, complexity, initial cost, security and privacy, health concern). We focus on application-oriented parameters in Table 2.

### 2.2. Structuring of Indoor Positioning Systems and Potential Traceability Technologies

Table 1 helps us to choose the right technology in the case of layer 0 and 1. Now we focus on the applicability of the RTLS. We found a multilevel selection criteria [20], where the three levels are Economic, Technical and Implementation. We made an overview of the most relevant RTLS technologies (excluding the no-radio based technologies) in Table 2. It summarizes these technologies with critical performance criteria, including accuracy, power consumption and costs.

Figure 3 gives us a classification of RTLS. We considered the left side of the figure because there are very few examples of no radio-based technologies. In the production environment, ultrasound signal transmission is also accompanied by a radio frequency (RF) pulse to combine the high accuracy of ultrasound with the high communications capacity of RF, which enables tracking of hundreds of simultaneously moving tags [21]. The technology is not used independently in the manufacturing environment because of the communications capacity, and environmental noise can degrade the localization accuracy [10]. One standalone application of RTLS is tracking the locations of construction resources such as labor, materials, machinery, and vehicles [22]. This application uses WiFi-based RTLS because GPS is limited in indoor environments, such as tunnels and buildings under construction. Another example is the development of a self-governing mobile robot navigation system for indoor construction applications [23]. Several navigation strategies with a mobile robot were tested with various combinations of localization sensors, including wheel encoders, sonar/infrared/thermal proximity sensors, motion sensors, a digital compass, and ultra-wideband (UWB) technology. The findings can be adapted to several potential construction or manufacturing applications such as robotic material delivery, inspection, and onsite security. Two RTLS applications of UWB and ultrasound technology have been tested in the SmartFactory KL [24].

Compared to other technologies, Zigbee has not spread substantially in industry [26]. However, industrial applications in the literature, such as the Zigbee positioning system for coal miners [27], have also been studied [28]. Laser-based systems are also used for navigation [29] and production tracking [30]. RFID is used in the production independently as an identification system [31] because only the presence of tags, such as barcodes, can be accounted for at the RFID reader. Other technologies must be used for real-time location [32]. An RFID-based RTLS solution exists, but this solution is less widely used because it is more expensive and inaccurate than UWB [33]. RTLS must be able to locate, track and identify objects in an indoor environment; therefore, RFID technology is not appropriate for RTLS. The literature in this area is controversial, so we consider various types of technology, such as BLuetooth, WiFi, Zigbee and UWB, as having the ability to support RTLS. Apple proposed the iBeacon protocol [34] in 2013 as another tool [35]. The new iPhone 11 from Apple already includes UWB beacons, which may be suitable for indoor positioning [36]. Moreover, it is essential to mention 5G technology [37], which could be crucial for future smart manufacturing, including highly accurate indoor localization. Because of the large-signal bandwidth and beamforming capabilities, localization and tracking could be more robust and efficient [38]. The technique is only now beginning to spread in industry; currently, it can only be applied in test and development environments [39]. The different indoor positioning-based traceability technologies can be combined in RTLS (already mentioned hybrid technology) to take advantage of different solutions in one system; for example, the ZigBee and UWB technologies or RSS measurements and a fingerprinting location algorithm usage for better position estimation [40]. To use hybrid technology, a platform is also needed. In general, roughly five layers are worth defining. A hardware layer, where position data are generated; a processing layer, where the position is calculated and filtered; a data layer, where location data is stored; a service layer, where we can optimize the system; a visualization layer, where we can analyze and monitor real-time location data.

In the next section, we describe potential industrial applications, while in Section 4 we propose a workflow of setting up an RTLS-based manufacturing support system. Finally, we describe a use case to illustrate the applicability of RTLS.

## 3. Industrial Applications of RTLS

An approach to RTLS selection is reviewed [16] and the RTLS based articles and the main advantages are summarized within Table 3 with the related fields of application in production and logistics categorized according to areas of use such as Quality Management, Safety and Efficiency Monitoring. Within the topics, reference is made to existing solutions, but possible directions for development are also presented in the following subsections.

Different application possibilities exist in manufacturing departments with RTLS. Table 4 presents various types of applications, where the RTLS-provided information is defined for every industrial application. The possible benefits define how we can validate the efficiency of the RTLS project. The applied positioning system provides real-time information about where equipment, semi-finished or finished products and specified logistic vehicles or workers are located in the manufacturing area [63].

### 3.1. Production Control with RTLS

Cycle time optimization is a critical task, especially in the case of modular or just-in-time (JIT) production [51]. A positive correlation exists between the potential of RTLS and JIT manufacturing. Several objects are defined as the main focus for tracking in the case of industry applications (mobile assets, workers, materials, key components, forklifts, pallets) [20].

The cycle time optimization is also possibly based on the position of products. For that, we need to know which products being produced are on the assembly line at every moment of time [51]. Pairing a semi-finished product with a tag makes inter-manufacturing tracking possible and makes the following information available:Time spent on the workstation for a given product;The production sequence;Which products are/have been on rework;Which products are/were in quality assurance;Average lead time for a particular product type (tact time);The goods in production are available with a continuous, real-time production status that supports production programming and shift design.

A position data-based decision-making approach is presented that relies on advanced data analytics for asset location systems to help production [52]. A potential use case in construction is discussed in Reference [68] and another application of material tracking in a pipe spool fabrication shop in [60].

Similar RTLS-based position data are already available in the literature, where a real-time connection between operator performance and varying product complexity was designed [55]. Another paper proposes an RTLS-based solution for a logistics problem with hybrid traceability technology (WiFi with RFID) to realize materials tracking, which can automate considerable amounts of warehouse work, such as stock-taking and storage positioning and checking [63]. Related research proposes an RFID-based intelligent decision support system architecture to handle production monitoring and scheduling in a distributed manufacturing environment [53]. Furthermore, RTLS technology can even be an element of reconfigurable facility layout planning. With its help, the processing steps of the activity and their relationships can be easily mapped and recorded in a database. More complex material flow can be provided (with information of the real flow between machines or congestion phenomena), which is not possible with a simple flowchart [56].

With the proliferation of Industry 4.0, it can be seen that there is also a strong emphasis on production monitoring. This is evidenced by the numerous references presented in the topic of production management.

### 3.2. RTLS in Logistics

Logistics efficiency is largely dependent on the movement of forklifts, pallet trucks and stacker trucks. For improving the logistics efficiency, we should know our current processes to see the points where lead time reductions can be achieved in the supply chain. A properly selected RTLS technology can be a tool for exploring logistics processes [61]. The following information is made available with logistical vehicles tracking:Routes and time spent in specific areas;Speed of forklift;Data for predictive maintenance;Forklift overall equipment effectiveness (OEE).

The objective of [58] is to propose an IoT and advanced data analytics-based warehouse management system (WMS) to enable smart logistics for Industry 4.0. The proposed IoT-based WMS can improve warehouse productivity, picking accuracy and efficiency, and it is robust to order variability. In [67], the authors present a sophisticated algorithm for tracking production and determining the traceability of a product. Reference [62] gives an example of how to apply RTLS across the supply chain and manage various assets within shop floors. A forklift based use-case is described in [66] where movement inside a warehouse determined by the RTLS is associated with assets that it picks up (attach through UHF-RFID reading) or puts down (detach through the loss of RFID signal).

IR-UWB-based RTLS has been deployed in an in-operation warehouse to track forklifts [75], and an RFID-enabled positioning system in AGV for a smart factory has also been presented [69]. Observations and lessons from simulation and testbed studies could be used to guide automated logistics within a smart manufacturing shop floor. The framework of an R-AGV-based material distribution scheme is proposed [73] based on an RTLS platform and electronic workshop map. The analysis and experimental results indicate that the R-AGV-based material distribution system provides new levels of process visibility and efficiency compared to traditional AGV-based distribution systems.

Tracking of transportation device is mandatory to obtain an accurate picture of intralogistics processes. A related paper describes an industrial forklift tracking problem that requires precise internal positioning [74]. It presents a study on the feasibility of meeting this challenge using UWB technology. Placing two tags on the forklift enables even more robust localization, as the measurements from the two tags are combined.

Based on the studied articles, it can be seen that much RTLS-related research is being done in logistics; there are still, of course, unexplored application possibilities.

### 3.3. Applications in Quality Management

It is possible to see where losses are generated with a real-time tracking system. The monitoring of the material flow with RTLS and the average duration of the processes provide information about problematic weak spots in the production process. Based on this information, possible reasons for the delay in production is discussed in Reference [76]. Root cause analysis is an essential component of quality assurance for the customers of the manufacturing company. After exploring the root cause, different action plans can be implemented, such as a review of the workflow, redesign of the workspace, education of workers and modification of work instructions.

Another possible advantage is RTLS-based dynamic work instruction. A crucial point in non-automated and human resource-required production processes is to ensure the well-supported work of the operators. One possible solution is showing just the information required to process the actual workpiece and no more. Based on the full traceability, we can show that the actual work instruction at every workstation based on the product information (from RTLS). There are relatively few practical applications on the topic, but potential development opportunities can be clearly identified such as dynamic work instruction.

### 3.4. RTLS for Safety

Collision avoidance is one possible improvement in the EHS (Environment, Health and Safety). Real-time alerts and notifications can be developed to prevent accidents based on the movement of vehicles and workers. A significant portion of the workplace accidents can be traced back to failure to use the required protective equipment. The monitoring of the personal protective equipment (PPE) usage is also an available function using RTLS [78]. RTLS technology can be used to control access to restricted areas for employees by sending automatic alerts whenever someone enters an unauthorized area [79]. Moreover, in the event of an emergency or natural disaster, such technology can be used to determine if everyone has already left the area or whether every worker used the designated route to leave the building [80].

The main purpose of contact-tracking solutions in industry is to help identify the contact matrix when the infection is recognized. RTLS is one possible solution to support to explore the potentially infected people [81]. Therefore, it can be used for the protection against the COVID-19 pandemic [85]. RTLS can be used in production systems also to monitor adherence to distance requirements between the operators. This is a possible useful function for a COVID-19-like epidemic situation. Due to the pandemic, the field of research for RTLS is topical, but there are also several solutions in the literature in the field of EHS (Environment, Health and Safety).

### 3.5. RTLS-Based Efficiency Monitoring

A related research article presents an RFID-based RTLS solution for performance metrics through RTLS data analysis to evaluate workflow performance and to obtain a lean process [82]. Spaghetti diagrams (visual representation using a continuous flow line to trace the path of an item or activity throughout a process) are time-consuming and static and, therefore, do not reflect the dynamics of logistics systems. RTLS was proposed to overcome this drawback [83].

The efficiency of a human resources personalized measurement is challenging. To achieve set goals, every organization must devise adequate, effective and efficient means of managing its HR. Related research reported on the development of an RFID and RTLS-based real-life personnel monitoring system to accurately and reliably estimate distance and coordinate the location of personnel at any instant [84]. This method can be used to measure—based on RTLS positioning data—how much time each product has spent at a particular station. This measurement can be further developed, and performance indicators can be obtained for workstations where operators work. By breaking down the overall process into sub-processes, we can refine the zones that allow for personalized performance tracking. The proposed RTLS can provide a solution to compute availability, a key parameter of OEE (Overall Equipment Efficiency) based on position data. Position data could improve the accuracy of the measurement of human resource efficiency (HRE) [86] and integrate other sensor measures for real-time activity monitoring [55]. Real-time sensor data assigned to the location of tools or machines on the shop floor allows online efficiency monitoring and supports the development and maintenance of digital twins [86] or intelligent decision-making systems.

In terms of human resources, there is still quite a bit of related research. More potential RTLS applications—like the mentioned personalized performance tracking—can help to achieve more efficient operations.

### 3.6. RTLS for Collaborative and Operator 4.0 Solutions

The future of manufacturing will be the personalization, and Industry 5.0 defines by the co-operation between man and machine [87]. During the current fourth industrial revolution, companies have realized they need to put humans back into industrial production with collaborative robots [88]. The workers need to be upskilled to provide value-added tasks in production to handle the mass customization and personalization for customers. This philosophy overlaps with the Operator 4.0 concept [89].

Shop floor trackers are one of the required technologies of Industry 5.0 [88]. RTLS could be a suitable solution for the full traceability on the shop floor. The smarter operator is an element of Operator 4.0 methodology [90], which is used to be the intelligent personal assistant-based solution. Real-time position data providing precise location information helps the system to make better decisions for operators and make possible the trajectory prediction of operators [91].

### 3.7. Analysis of Position Data and Building Data-Driven Solutions

Raw data provided by the RTLS cannot directly be utilized to support the manufacturing. The purpose of this section is to introduce data based solutions and the related data analysis techniques needed for data pre-processing and building data-driven solutions. Again, a systematic examination of literature in Scopus was done, using the keyword set (“indoor positioning” AND (“machine learning” OR “data science” OR “data mining”). The network of the mentioned keywords can be seen in Figure 4. Based on this network we can define the key thematic group of machine learning techniques and the related application areas that will also be discussed in detail in this section.

Data mining techniques are reviewed in Reference [92] to solve indoor navigation systems problems. The performance of the RTLS is shown via the integration of different features and classification algorithms, including decision tree, multi-layer perceptrons, and Bayesian networks [93]. In another article, naive Bayes theorem-based classification techniques and other classification techniques to enhance the classification accuracy are compared to identify the best location estimation algorithm [94]. K-nearest neighbor [95], support vector machine [96], decision tree [97], naıve Bayes [98] and Bayesian network methods [99] are compared and combined with ensemble learning algorithms to improve the performance, i.e., accuracy, f-score and computation time [100]. Decision tree-based classification is applied to estimate the position to improve the accuracy [97]. The clustering machine learning (ML) technique is usually used to improve RTLS accuracy, like K-means clustering backpropagation NN [101], Spatial Division Clustering (SDC) method [102], affinity propagation clustering [103]. The feeding behavior of cows is measured with RTLS in [104]. The presence at the feeder (feeding probability) of the cows was calculated using the logistic regression model. Support Vector Regression (SVR) is used to calculate the efficient RTLS [105]. Based on the aforementioned literature study, the common combinations of ML techniques are presented in Table 5 and RTLS technologies can be seen in Figure 5. Based on the qualitative analysis of the literature it can be highlighted that many researchers have successfully applied NNs to the indoor positioning problem via convolutional neural networks (CNNs) [106]. A ZigBee [107] indoor positioning research scheme based on the location fingerprinting approach uses an NN locating model. This model, with the signal-index-pair data pre-processing method, is used to increase positioning precision [108]. Related research uses a particle swarm optimization-based backpropagation (PSO-BP) NN to determine the relationship between RFID signals and the position of a tag for an RFID-based positioning system [109]. Furthermore, to improve the quality of training samples, the experimental data are pre-processed via Gauss filtering.

The following section is devoted to show how RTLS systems and the presented models can be integrated into one system.

## 4. Steps of Setting Up an RTLS for Manufacturing Support

An installation of RTLS is described in this section with a proposed workflow to illustrate the difficulties of RTLS projects. Figure 6 shows the necessary steps for RTLS-based process analysis. In general, the first step of an RTLS based digitization project is the identification of the requirements, where we define the physical area on the shop floor and the possible applications. The next step is the installation of the sensor network. After the system is running and the position information is being successfully gathered, the accuracy of the system should be validated. The multi-tag concept substantially improves the object detection probability and makes the system more robust [111,112]. Generally, the position engines of RTLS apply filtering methods to pre-process the position data (e.g., a Kalman-filter in the case of GPS) [113]. Several accuracy improvement solutions are available based on RTLS, including regression [114] and k-nearest neighbor classification [95].

With the spread of RTLS, position data pre-processing and cleaning methods have become an important research topic, based mainly on pedestrian dead reckoning [115] and wireless signal positioning methods [116]. The cleared and filtered position data provides more accurate information to the production system.

The integration of position data into the Manufacturing Execution System (MES) is a crucial element of the implementation.

To obtain usable data from the position information, zones of the manufacturing process should be defined. A zone (Station ID) represents a workstation or storage space, and the RTLS can obtain zone information from every tag in real-time. Figure 7 shows the connections among the RTLS, MES and production. At the beginning of production, the operator pairs a product with an RTLS tag (with a barcode scanner or manually at a PC) to identify the actual product ID in the system. The RTLS provides the zone information (based on the position data of tags and zones definition) with a timestamp to the MES with an application programming interface (API). The MES changes the status of the actual product ID based on the information from RTLS (e.g., the product is tested at a testing station). If a digital interface is provided at the workstation, then the MES can show the work instruction for the actual product or can set the optimal cycle time based on the product content [51].

## 5. Analysis Based on Position Data—A Case Study

Position data-based production tracking has considerable potential to optimize production processes. In this section, we present a case study based on an implemented RTLS in a manufacturing environment. The purpose of the position data in production is to transform to relevant information, in the interest of comparing the defined production zones with the position based clusters. Hidden information can be extracted from the position data for production management.

Our use case is an anonymized example from a Tier 1 supplier company from the automotive sector. The production company used the Sunstone-RTLS Ltd. system, which is accurate to 50 cm with eight anchors per every 2000–3000 m${}^{2}$. The system architecture is shown in Figure 8. There are seven workstations that are used to produce a small wire harness. The zones define the workstations, storage units and routes. The workstations are Tubing station I., Tubing station II., Channeling station, Test station, Screwdriver station, Packaging station and Quality check. The operator attaches the RTLS tag to the product at the first station (Tubing station I/II), and the final station is the Packaging.

The goal of the position-based zone identification is to determine the temporary storage at the temporary station area: a K-means algorithm is applied for position data classification. Figure 9 shows the classified position data where the algorithm detects the three small storage areas over the pre-defined zones. These three undefined or unplanned stations could be the cause of several losses.

Thanks to the position data, the full traceability of products is available. The cycle time of workstations can be measured based on the classified zone data. The operator scans the product identification label at the first workstation, where the system paired it to the current tag ID (ID is also scanned at the station). The spent time of the actual *p* product ($T_{p}^{z}$) in the actual zone (*z*) is the difference between the last timestamp ($T_{p}^{z}\left(l\right)$ of the actual product (*p*) position data in the actual zone and the first one ($T_{p}^{z}\left(f\right)$).

Figure 10 shows the cycle time deviation of every workstation. The boxplot shows the distribution of the times related to the production of more than 150 products (in one shift). We can notice that the packaging station has less cycle time and also the smallest deviation. The reason is, we cannot identify the real finish time at this station due to the fact that they collect the tags traced by the RTLS at this station after the production, and instead, the operator scans the test label during the packaging. The figure shows the Tubing station I. is the bottleneck, but we can see in Figure 9, there is a second station (Tubing station II.), which is a spare workstation. In the current situation, there are not enough resources to operate both workstations, but it could be the solution to improve the process.

As this example highlighted, the RTLS can provide accurate and real-time information about the current status of the production process that could be utilised in the development of production processes.

## 6. Conclusions, Limits and Future Direction of Research

The purpose of this article was to provide a comprehensive overview of the application and development possibilities of RTLS in the manufacturing field. The overview of the solutions determined the value of the positioning data and specified which traceability technologies are suitable for real-time locating in different situations to ensure traceability. Our research explored the possible applications in the production and logistics process. Finally, the implementation of RTLS and a data cleaning method are represented. The end of the article presented a case study, in which we demonstrated what kind of information an RTLS system can provide.

The research pointed out that information extracted from RTLS is highly applicable for performance monitoring. Based on this fact, RTLS supported LEAN projects are very important research topics of the future. This article also introduced that machine learning and state estimation techniques are getting used more and more widely in the development of position data based models. Another conclusion of the case study is that typical states of the production process can be easily determined based on clustering algorithms. Analyzing the sequence of these means a significant increase in the understanding of the processes and in support of process models. According to this we believe that process mining is the most relevant research topic for the future.

It is also important to highlight that an installed RTLS makes the integration of more sensor data available, and in this way the quick implementation of IoT solutions. We believe that this opportunity is beneficial primarily in the development of existing processes, like the introduction of brown field Industry 4.0 solutions.

The advantage of these solutions can be exploited well if the production system is supported by a Manufacturing Execution System (MES), in which a system ensures that information derived from position data can be used in production process optimizations. In line with this, an RTLS project should be connected with an MES development. This MES development process can be supported by semantic models, which are helping to structure sensor and production data. The application of these models is also a research and development topic for the future.

## Figures and Tables

**Figure 1 sensors-20-06766-f001:**
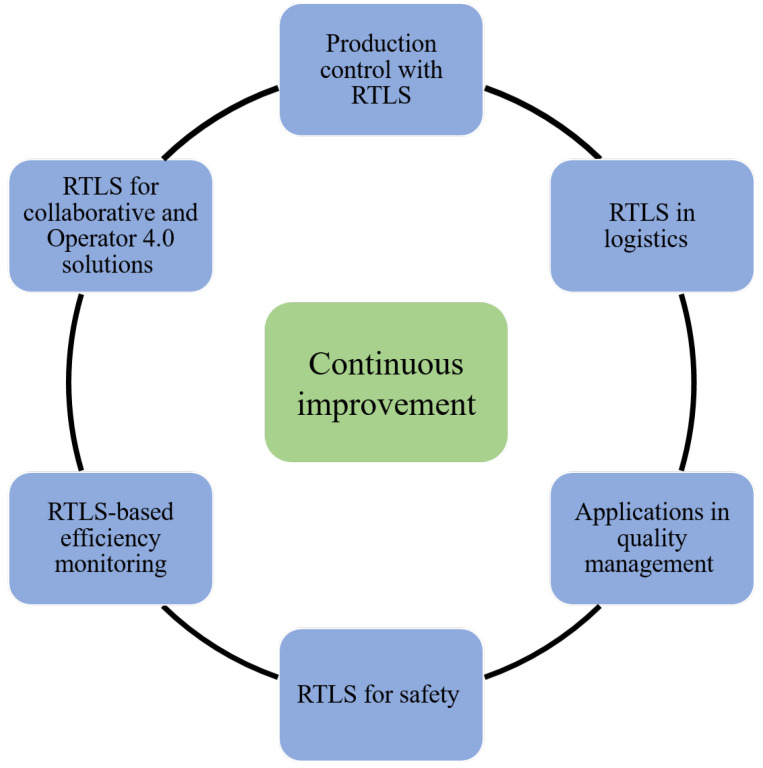
Use of real-time locating system (RTLS)-based positioning information by the different parts of the manufacturing environment. Continuous improvement is a central element of an RTLS project.

**Figure 2 sensors-20-06766-f002:**
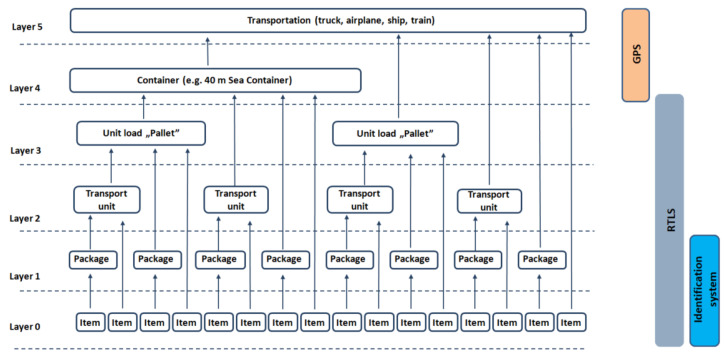
Identification levels in a production system. Layers define the logistic units from raw material (items) to trucks (transportation).

**Figure 3 sensors-20-06766-f003:**
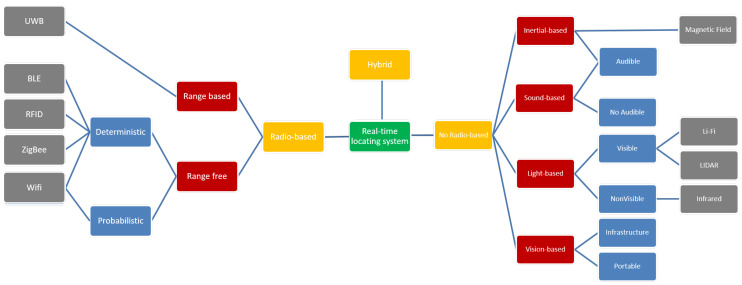
Classification of RTLS [25].

**Figure 4 sensors-20-06766-f004:**
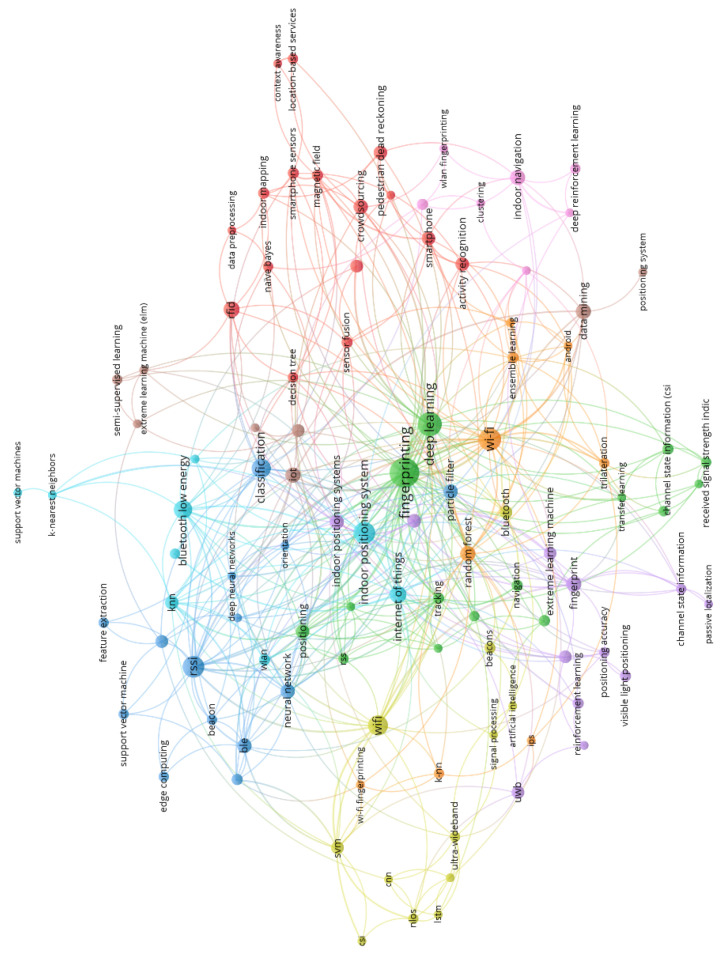
Network of keywords based on Scopus database.

**Figure 5 sensors-20-06766-f005:**
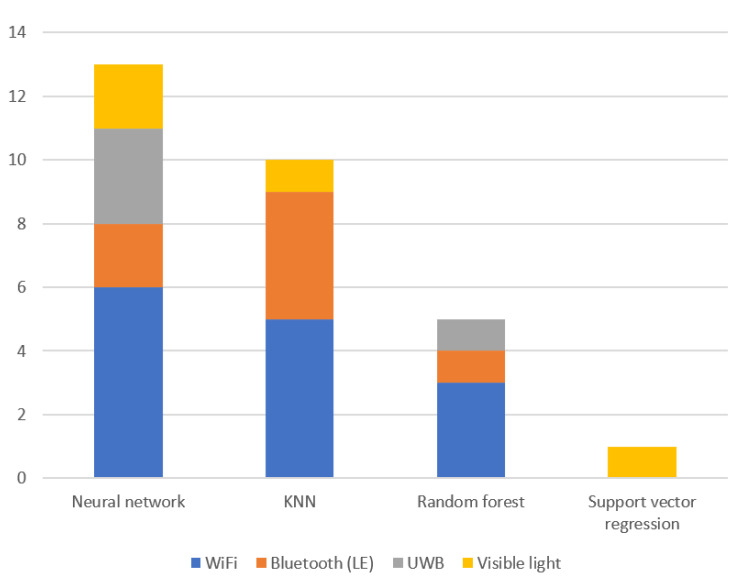
Concurrence of machine learning (ML) techniques and RTLS technologies in articles.

**Figure 6 sensors-20-06766-f006:**
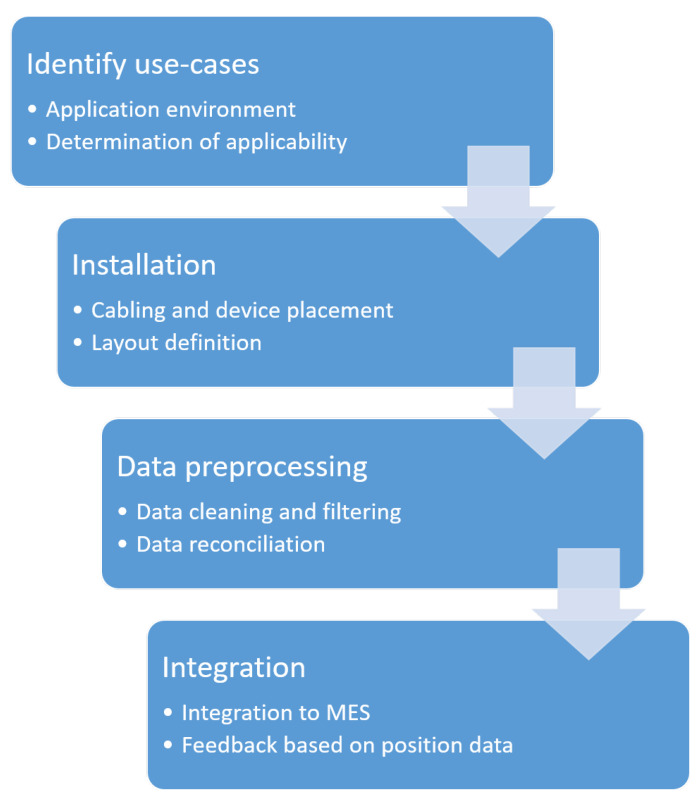
The full setup of RTLS in manufacturing. After the physical system installation, the layout and zone definition is necessary for system integration into the Manufacturing Execution System.

**Figure 7 sensors-20-06766-f007:**
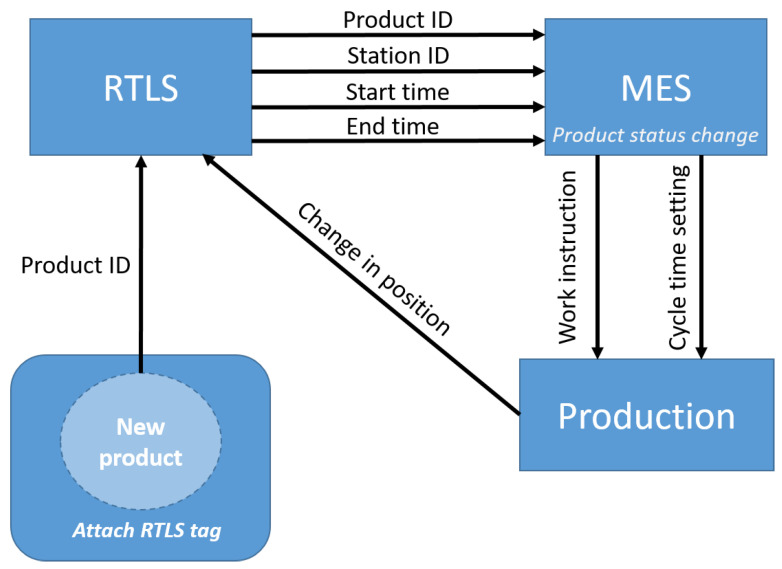
The real-time connection between the Manufacturing Execution System (MES) and production is available based on the RTLS.

**Figure 8 sensors-20-06766-f008:**
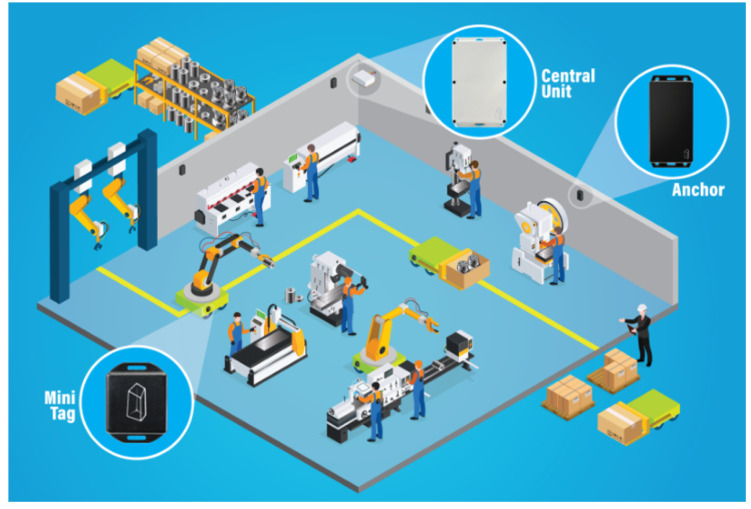
The infrastructure of the Sunstone-RTLS. Every central unit (CU) has eight anchors (which collect data from tags), and the CUs can be connected to create a cascade installation.

**Figure 9 sensors-20-06766-f009:**
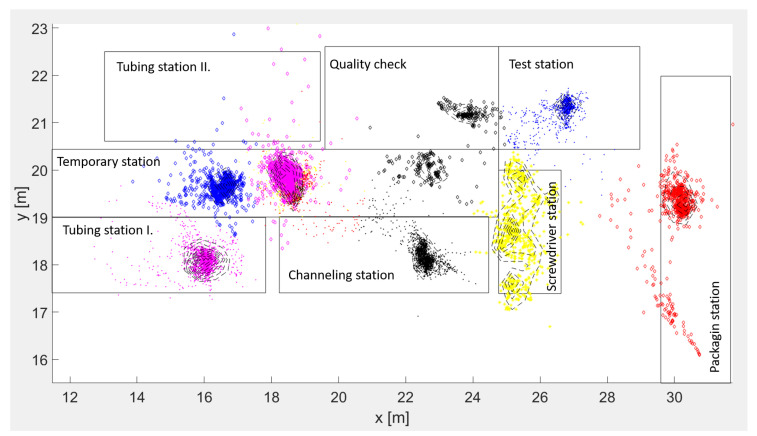
The production layout with seven pre-defined workstations. The classified and pre-defined (rectangles) zones are shown. The algorithm detects three small areas behind the Temporary station and indicates that Tubing station II was not used in this period.

**Figure 10 sensors-20-06766-f010:**
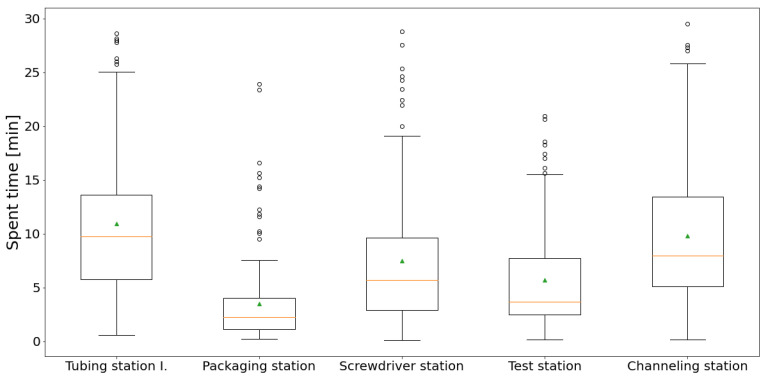
Boxplots of the cycle times measured at different zones of the production process. (Green triangles represent the averages, while red lines the medians).

**Table 1 sensors-20-06766-t001:** The most widely applied solutions for traceability. The advantages and disadvantages help us to choose the right solution for the right layer (see in Figure 2).

Solution		Advantages	Disadvantages
Identification system	Barcode	Cost effective, spread technology	The potential for human error is high
			Additional installation cost per every layout changing
		The cost per item is low	The cost of printed labels can be relevant in the case of enormous stock
	RFiD	Cost effective, spread technology	Every new station needs new hardware and installation
		The cost per item is low	Additional installation cost per every layout changing
		The human error can be minimized	
GPS		Spread technology	The accuracy is not enough in the case of indoor positioning
		Many devices are already compatible	The reliability is low in the case of indoor positioning
		It is highly scalable	
RTLS		The error of human is excluded	The cost depends on the number of tracked items
		The traceability is available at the covered area	
		The system is fully flexible, any layout changing can be handle in the software application	Any new item can be added at any time to the system (highly scalable)

**Table 2 sensors-20-06766-t002:** Review of indoor positioning-based traceability technologies.

Techn.	Tag Cost [41,42,43]	Module Cost [44]	Accuracy	Space dim.	Power cons. [44,45]
Scale	L:<3$ M:<10$ H:>20$	L:<10$ M:<40$ H:>70$	L:>1 m M:10 cm H:<10 cm	2D/3D	L:<100 mA H:>200 mA
Zigbee [46]	M	M	M	2D	L
RFID [47]	L	H	L/M	2D/3D	L
BLE [48]	L	L	L/M	2D	L
Wifi [49]	H	H	L/M	2D	H
UWB [50]	H	H	H	2D/3D	H

**Table 3 sensors-20-06766-t003:** Industrial applications of RTLS technologies.

Application Type	Application	Technology
Production	Cycle time optimization	UWB [51]
Production	Position data-based decision making	UWB [52]; RFID [53,54]
Production	Activity-Time monitoring in production line	UWB [55]
Production	Digital Facility Layout Planning	Independent [56]
Logistics	Logitics management	RFID [57];Hybrid [21]; Independent [20]
Logistics	Warehouse management	RFID [58]; WiFi [59]
Logistics	Pallet management	RFID [31]
Logistics	Material/component and production tracking	WiFi [60]; UWB [31,61]; RFID [31,62,63,64,65]; Hybrid [66];Laser [30];Barcode [67]
Logistics	Assets tracking	Bluetooth [68]; RFID [69,70,71,72]; Hybrid [73]; UWB [74,75];Laser [29];Barcode [67];ZigBee [46]
Quality	Weak spot analyzis in production	UWB [76]
Safety	Safety management	RFID [33]
Safety	Collision avoidance	UWB [77]
Safety	Personal protective equipment monitoring	Hybrid [78]
Safety	Person tracking	ZigBee [27]; RFID [79]; UWB [80]
Safety	Contact tracking	Independent [81]
Efficiency monitoring	Performance of manufacturing process	RFID [82]
Efficiency monitoring	Lean manufacturing	UWB [83]; BLE [48]
Efficiency monitoring	Human resource monitoring	RFID [84]

**Table 4 sensors-20-06766-t004:** Application of RTLS in manufacturing, the useful information it provides and possible benefits.

Application Name	Information Provided by RTLS	Possible Benefits
Production control with	Footprint of semi-finished	More efficient production planning
RTLS (Section 3.1)	products and cycle time control	
RTLS in logistics	Tracking of logistical assets	More cost-effective logistics process planning
(Section 3.2)	in the production system	
Applications in quality	Root cause analysis depends on position data	Help quality management department
management (Section 3.3)		comply with standards and regulations
RTLS for safety	Human and material handling equipment	Reduction in occupational accidents
(Section 3.4)	tracking can help in collision detection	
RTLS-based efficiency	Efficiency indicators provide a realistic	Real-time efficiency monitoring assigned to machines
monitoring (Section 3.5)	picture of real-time production	or tools can support making better decisions
RTLS for collaborative and Operator	Precise real-time position of operators to	More efficient decision making
4.0 solutions (Section 3.6)	predict the possible collaboration situations	for the smart operator and collaborative system

**Table 5 sensors-20-06766-t005:** Data mining techniques and areas of RTLS-based application.

Method	Definition	Data Analytic Techniques	Application Areas	RTLS-Based Applications
**Classification**	Discriminating data into different labeled subsets pursuant to class attribute. Retrieving important and relevant infor- mation about data and metadata.	Neural network Support vector machine (SVM) Decision tree k-Nearest neighbor Bayesian network Genetic algorithm	Pre-defined distribution (e.g., identification of differences) Fault detection Anomaly detection problems	For intralogistics navigation problems [92], shows the performance of RTLS [93], find the best location estimation algorithm [94]
**Clustering**	Grouping the database according to their similarities. Discovering similarities and dissimilarities between the data.	Partition based algorithms (e.g., K-means, fuzzy c-means) Hierarchical clustering (e.g., dendrogram) Density-based method Grid-based methods Model-based methods	Data segmentation (division into homogeneous sets) Identification of typical prototypes (e.g., simultaneous identification of time-homogeneous periods and their averages/trends)	Improve RTLS accuracy [101,102,103] Pedestrian motion learning [110]
**Regression analysis**	Identifying and analyzing the relationship between variables. Predicting and forecasting the process or dependent variables.	Multivariate linear regression Neural network Regression tree	Creating a model that predicts time (e.g., creating a model for predicting temperature)	Used to calculate the efficient RTLS [105] feeding behavior of cows [104]

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
