# Peer review of "Real-Time Locating System in Production Management"

_sensors, 2020, doi:10.3390/s20236766_

Round 1
Reviewer 1 Report
This paper aims to review the potential manufacturing applications of RTLS. It does so by using a comprehensive and adequate literature review.
Overall this is a good paper well written and structured. The main issue with the paper is that the introduction could offer a better job of situating the problem and explaining why RTLSs are important, and what are the implications of a wider use of this techniques.
I think that a case-study is not provided as claimed in the conclusion section. It is only a little example. Please amend this issue.
Maybe a limits sections and future directions could improve this manuscript.
I miss also a section about the RTLS-human interaction in the manufacturing sectors, addressing in this sense also opportunities, difficulties and challenges.
As a minor comment, please avoid RTLS acronym in the abstract section.
Author Response
Reply to reviewer I. - # sensors-983781
Title: Real-time locating system in production management
Sensors special issue: Indoor Positioning and Navigation
Dear Reviewer,
We are grateful for your constructive critique and useful comments that we received in the first round of the review process. We have addressed all the comments as explained below. We still hope that the modifications have significantly improved the quality of the manuscript.
In thi revision round, we tried our best to improve the paper further and provide more details about how the paper has been improved based on your comments.
The article was significantly extended based on editorial recommendation, in which extension the comments of reviewers regarding the overview of literature was considered. Based on these notes, we introduced the research more in details, including the search methods. A summary of these methods was also included in the article, as well a visual introduction of the network of keywords used during the literature examination. In this extension, our colleague László Bántay was heavily involved, from that reason we kindly ask to add him to the list of authors, as co-author.
Overall this is a good paper well written and structured. The main issue with the paper is that the introduction could offer a better job of situating the problem and explaining why RTLSs are important, and what are the implications of a wider use of this techniques.
Many thanks for this useful comment. Based on it, the Introduction was extended with a section, emphasising the importance of RTLS in management and development of production systems, in a nutshell, the potential what motivated our research.
I think that a case-study is not provided as claimed in the conclusion section. It is only a little example. Please amend this issue.
Many thanks for this useful remark. The case study in Section 5 was extended in order to give a complete picture of application example, in line with that, we also amended the related sentence in Conclusion.
Maybe a limits sections and futtre directions could improve this manuscript.
We are grateful for this suggestion, in line with that we significantly extended the summary, we also gave a new title to it, which reflects the amended content.
I miss also a section about the RTLS-human interaction in the manufacturing sectors, addressing in this sense also opportunities, difficulties and challenges.
We are grateful for this important remark! We inserted a subsection called “RTLS for collaborative and Operator 4.0 solutions” in the article. In line with the suggestion, Section 3.4 was also amended.
As a minor comment, please avoid RTLS acronym in the abstract section.
Thank you for your remark, we corrected it in abstract!
Reviewer 2 Report
The topic of the paper is interesting. I think there is some potential, but the authors have to carefully think about the structure of the paper, which now is very very confusing. Please consider which is the aim of the paper and “build” the paper in this direction. Each section should add some information useful for the aim of the paper. Moreover, the headings of the sections should be more descriptive and coherent with their contents. It also important to well specify what has been found in the existing literature and what is a proposal of the authors.
- Introduction
- Since at the beginning of the paper you say that you consider IPS and RTLS as synonyms, I suggest to use only one of the two terms in your manuscript. I think this can help the reader in not making confusion.
- Levels of location information in manufacturing industries
- I think this section is a bit confusing and not well structured. I think there are too many (interesting) information, put all together, without an appropriate deepening. Please consider to restructure this section and to give more details about the information you are writing: simply giving a reference to a paper could not be enough if you don’t explain what this reference includes and why you consider important to cite it. Moreover, I would consider to put some sub-sections to better divide the different concepts.
- Although I understand the aim of Table 1, it is not clear to me if the comparison proposed there is meaningful. While Barcode, RFId and GPS systems refer to specific (and clear) technologies, RTLS can include different systems, which can use different technologies and devices. I think it is hard to derive general directions about RTLS without referring to a specific system. If this is not the case, please specify in the description of the table.
- Potential industrial applications of RTLS
- In this section it is not clear what has been derived from previous researches and what is instead proposed or derived by the authors. you are proposing something new, you have to explain on what your considerations are based (for example, experience, seen in practice etc.)
- For example, in Table 3: why are you proposing these “possible benefits”? where are they coming from? Since they are not all (you say they are “possible”), are you sure these are the most important ones?
- For the existing literature, you have to specify how you searched it, to demonstrate that your review is complete. For example, I don’t see the following papers:
Zuin S., Calzavara M., Sgarbossa F., & Persona A. (2018). Ultra Wide Band Indoor Positioning System: analysis and testing of an IPS technology. IFAC-PapersOnLine, 51(11), 1488-1492.
Peron, M., Fragapane, G., Sgarbossa, F., & Kay, M. (2020). Digital Facility Layout Planning. Sustainability, 12(8), 3349.
And I am worried you could have missed many others.
- Supporting solutions and technologies
- I don’t fully understand the title of the section of the link to its contents. I also don’t understand why you are talking about your case here: probably it is better to dedicate only one section where you explain it completely.
- Since you have many references about industrial cases, I would structure a summarizing table showing all the cited references, classifying them according to the technology used, to their field of application etc.
- Analysis based on position data
- Again, I don’t fully understand the title of the section of the link to its contents.
- It is also confusing to find general information and information related to the case study.
- Why have you considered this practical example useful for this paper, especially for the aim of this paper? Please specify and support why this case is representative for this paper.
Author Response
Reply to reviewer II. - # sensors-983781
Title: Real-time locating system in production management
Sensors special issue: Indoor Positioning and Navigation
Dear Reviewer,
We are grateful for your constructive critique and useful comments that we received in the first round of the review process. We have addressed all the comments as explained below. We still hope that the modifications have significantly improved the quality of the manuscript.
In this revision round, we tried our best to improve the paper further and provide more details about how the paper has been improved based on your comments.
The article was significantly extended based on editorial recommendation, in which extension the comments of reviewers regarding the overview of literature was considered. Based on these notes, we introduced the research more in details, including the search methods. A summary of these methods was also included in the article, as well a visual introduction of the network of keywords used during the literature examination. In this extension, our colleague László Bántay was heavily involved, from that reason we kindly ask to add him to the list of authors, as co-author.
The topic of the paper is interesting. I think there is some potential, but the authors have to carefully think about the structure of the paper, which now is very very confusing. Please consider which is the aim of the paper and “build” the paper in this direction. Each section should add some information useful for the aim of the paper. Moreover, the headings of the sections should be more descriptive and coherent with their contents. It also important to well specify what has been found in the existing literature and what is a proposal of the authors.
We consider your suggestion very useful, in line with it the Introduction was amended with a new paragraph including our motivation, as well a brief review of the aim and message of the Section.
- Introduction - Since at the beginning of the paper you say that you consider IPS and RTLS as synonyms, I suggest to use only one of the two terms in your manuscript. I think this can help the reader in not making confusion.
Thank you for your remark. We changed all terms to RTLS.
- Levels of location information in manufacturing industries - I think this section is a bit confusing and not well structured. I think there are too many (interesting) information, put all together, without an appropriate deepening. Please consider to restructure this section and to give more details about the information you are writing: simply giving a reference to a paper could not be enough if you don’t explain what this reference includes and why you consider important to cite it. Moreover, I would consider to put some sub-sections to better divide the different concepts.
Many thanks for the useful comment. Section 2 is split into two subsections. In line with the suggestion content of the subsections were significantly extended.
- Levels of location information in manufacturing industries - Although I understand the aim of Table 1, it is not clear to me if the comparison proposed there is meaningful. While Barcode, RFId and GPS systems refer to specific (and clear) technologies, RTLS can include different systems, which can use different technologies and devices. I think it is hard to derive general directions about RTLS without referring to a specific system. If this is not the case, please specify in the description of the table.
We are grateful for your critical remark! Table 1 is modified based on your useful comments.
- Potential industrial applications of RTLS - In this section it is not clear what has been derived from previous researches and what is instead proposed or derived by the authors. you are proposing something new, you have to explain on what your considerations are based (for example, experience, seen in practice etc.)
Thanks for your comment, the Section was significantly extended, new subsections were included. At the end of each subsection, we made a summary of the relevant applications and related research directions given the article’s aim, the applicability in production systems.
- Potential industrial applications of RTLS - For example, in Table 3: why are you proposing these “possible benefits”? where are they coming from? Since they are not all (you say they are “possible”), are you sure these are the most important ones?
Many thanks for this useful question. The table and the related text was modified, highlighting that there are very few examples for widely industrial use of RTLSs. The prime purpose of our research was to discover what potential benefits and application options can be foreseen in which production systems using the related methods. These potential benefits are discussed in details subsection by subsection, and if there is a documented case study, we also reference it.
- Potential industrial applications of RTLS - For the existing literature, you have to specify how you searched it, to demonstrate that your review is complete. For example, I don’t see the following papers:
- Zuin S., Calzavara M., Sgarbossa F., & Persona A. (2018). Ultra Wide Band Indoor Positioning System: analysis and testing of an IPS technology. IFAC-PapersOnLine, 51(11), 1488-1492.
- Peron, M., Fragapane, G., Sgarbossa, F., & Kay, M. (2020). Digital Facility Layout Planning. Sustainability, 12(8), 3349.
And I am worried you could have missed many others.
We are grateful for the great article suggestions. We cited them. In order to avoid exclusion of any relevant article, a systematic examination of literature was performed following the PRISMA protocol. Details of the examination are discussed in the Introduction. The extended bibliography contains now more than 165 citations.
- Supporting solutions and technologies - I don’t fully understand the title of the section of the link to its contents. I also don’t understand why you are talking about your case here: probably it is better to dedicate only one section where you explain it completely.
This remark was also beneficial, based on it the description of the case study has been moved to the last (5th) section.
- Supporting solutions and technologies - Since you have many references about industrial cases, I would structure a summarizing table showing all the cited references, classifying them according to the technology used, to their field of application etc.
This comment is very helpful, in line with it an additional table was placed in Section3.1, where you can see, which technologies were used in which field of applications.
- Analysis based on position data - Again, I don’t fully understand the title of the section of the link to its contents.
We deeply regret our inattention, in order to eliminate the inconsistency we wrote a separate subsection for the opportunities in data analysis (see Sec.3.7 Analysis of position data and building data-driven solutions) and the case study was moved to another subsection.
- Analysis based on position data - It is also confusing to find general information and information related to the case study. Why have you considered this practical example useful for this paper, especially for the aim of this paper? Please specify and support why this case is representative for this paper.
The case study was extended in order to illustrate the methods suggested by us in the section before. The case study gives a good explanation of how to position data can be transformed into useful information by clustering. In the end, there is a summary, which has the purpose of showing the connection between details of the example and the suggested method.
Reviewer 3 Report
The article presents a topic that is current in the industry 4.0 since it presents the incorporation of real-time localization systems in all production processes.
In general, the article presents a wide variety of references of great interest, although on some occasions they are presented without being properly grouped and/or justified.
The first three sections: introduction, localization levels and potential applications are well presented and easy to follow. However, sections 4, 5 and 6 (solutions and technologies; data analysis; conclusions), which are fundamental to this article, have no connection with the rest and/or present very partial results. These sections should be completely rewritten. As suggestions for doing so I include the following:
Suggestions:
- Why do you start the section by including Figure 4 and not continue to exploit this or an equivalent one? It should be the thread.
- In the paragraph that begins in line 215, you start talking about UWB with TDoA. Why? At what point do you indicate that this is really the best technology? It is understood that this is a more generalist article, not something based on a specific solution that, moreover, was not justified before.
- There is talk of different techniques for the development of localization algorithms, where techniques such as Kalman, NN, fingerprinting... are mixed with technologies such as Zigbee. Again, the technologies should be clear in the preceding sections and in the algorithms, it would be good to classify them according to the possibilities they provide to any technology.
- Finally, in the integration section, it would be good to include in previous sections references to architectures and/or platforms that allow real-time localization with multiple technologies.
- The conclusions should be rewritten and greatly improved.
On the other hand, there are concrete details to consider:
- On line 8 (abstract), "and" is written twice.
- In line 26 several technologies are mentioned (infrared, ultrasound, ...), but there is no mention of RF which, by the way, is the only one considered in later sections.
- In the label of table 2, although it is understood, it is necessary to define L, M and H.
- To finish section 2, it would be important to show RTLS solutions that combine or can combine multiple technologies, thus taking advantage and/or coverage of each one. For example, we suggest reading this reference: https://doi.org/10.3390/s130202220
- In figure 3, although it corresponds to another reference, there is an imprecision in the techniques based on range. The location parameters (RSSI, Angle, Time) are put on the same level with technologies (UWB). In fact, the article talks a lot about technologies, but not so much about the physical parameters that they provide and therefore give rise to results as those seen in table 2.
- In section 3, when the items of lines 105-112 are presented, there is no mention of any of the items that appear in the subsections, such as safety. In addition, table 3 does not seem to have any relation to these items.
- In subsection 3.4 on safety, mention should be made of some PPE (Personal Protective Equipment) monitoring system, especially when COVID-19 is mentioned.
- The acronym MES is defined very late in the text (line 244), it appears in figure 4 and in the text a little earlier.
Author Response
Reply to reviewer III. - # sensors-983781
Title: Real-time locating system in production management
Sensors special issue: Indoor Positioning and Navigation
Dear Reviewer,
We are grateful for your constructive critique and useful comments that we received in the first round of the review process. We have addressed all the comments as explained below. We still hope that the modifications have significantly improved the quality of the manuscript.
In this revision round, we tried our best to improve the paper further and provide more details about how the paper has been improved based on your comments.
The article was significantly extended based on editorial recommendation, in which extension the comments of reviewers regarding the overview of literature was considered. Based on these notes, we introduced the research more in details, including the search methods. A summary of these methods was also included in the article, as well a visual introduction of the network of keywords used during the literature examination. In this extension, our colleague László Bántay was heavily involved, from that reason we kindly ask to add him to the list of authors, as co-author.
Why do you start the section by including Figure 4 and not continue to exploit this or an equivalent one? It should be the thread.
We are grateful for the useful remark, in line with it, the article was restructured, and components of Figure 4 were discussed more in detail.
In the paragraph that begins in line 215, you start talking about UWB with TDoA. Why? At what point do you indicate that this is really the best technology? It is understood that this is a more generalist article, not something based on a specific solution that, moreover, was not justified before.
We are very sorry for our inattention, the highlighted part indeed contains specific information, which was placed incorrectly in this section. This part was removed, the whole section is generalized, which means, that now steps of solution development based on indoor position systems are discussed independently from technologies.
There is talk of different techniques for the development of localization algorithms, where techniques such as Kalman, NN, fingerprinting... are mixed with technologies such as Zigbee. Again, the technologies should be clear in the preceding sections and in the algorithms, it would be good to classify them according to the possibilities they provide to any technology.
Many thanks for this useful comment, analysis techniques were separated from technologies. An independent subsection was made to discuss analysis methods.
Finally, in the integration section, it would be good to include in previous sections references to architectures and/or platforms that allow real-time localization with multiple technologies.
Thanks to your suggestion, this section was fully reworked.
The conclusions should be rewritten and greatly improved.
Thank you, Conclusion was rewritten and amended with the definition of applicability limits of RTLS along with further research directions.
On line 8 (abstract), "and" is written twice.
Thank you for this remark, corrected.
In line 26 several technologies are mentioned (infrared, ultrasound, ...), but there is no mention of RF which, by the way, is the only one considered in later sections.
Thank you very much, we extended the list with RF.
In the label of table 2, although it is understood, it is necessary to define L, M and H.
Thanks for the attention, the caption of Table 2 was amended with proper definitions.
To finish section 2, it would be important to show RTLS solutions that combine or can combine multiple technologies, thus taking advantage and/or coverage of each one. For example, we suggest reading this reference: https://doi.org/10.3390/s130202220
We are grateful for the suggestion; the section was modified according to it.
In figure 3, although it corresponds to another reference, there is an imprecision in the techniques based on range. The location parameters (RSSI, Angle, Time) are put on the same level with technologies (UWB). In fact, the article talks a lot about technologies, but not so much about the physical parameters that they provide and therefore give rise to results as those seen in table 2.
Thank you for the useful comment, based on it the figure, Table 2 and the related text is modified. The suggested method is also amended, highlighting the importance of choosing the right technique in light of circumstances of the application.
In section 3, when the items of lines 105-112 are presented, there is no mention of any of the items that appear in the subsections, such as safety. In addition, table 3 does not seem to have any relation to these items.
Many thanks for the remark, the related paragraph was modified in order to raise the coherency of the text and the table.
In subsection 3.4 on safety, mention should be made of some PPE (Personal Protective Equipment) monitoring system, especially when COVID-19 is mentioned.
We are thankful for this note; some PPE related ideas were added to the section.
The acronym MES is defined very late in the text (line 244), it appears in figure 4 and in the text a little earlier.
We are grateful for this important remark. We defined this in the figure label also and moved it in the first place where we mentioned MES.
Round 2
Reviewer 3 Report
Nothing.